Using DNA metabarcoding to assess insect diversity in citrus orchards

Liu Chenxi 1 liuchenxi@caas.cn
Ashfaq Muhammad 2
Yin Yanfang 1
Zhu Yanjuan 1
Wang Zhen 1
Cheng Hongmei 1
http://orcid.org/0000-0002-3081-6700 Hebert Paul 2
1 Sino-American Biological Control Laboratory, Institute of Plant Protection, Chinese Academy of Agricultural Sciences , Beijing , China
2 Centre for Biodiversity Genomics and Department of Integrative Biology, University of Guelph , Guelph, Ontario , Canada
Ayayee Paul
Electronic publication date: 2023 May 5
Publication date: 2023
Volume: 11
Electronic Location ID: e15338
Received 2022 Sep 23; Accepted 2023 Apr 11
Copyright: © 2023 Liu et al.
Copyright year: 2023
Copyright holder: Liu et al.
License: This is an open access article distributed under the terms of the Creative Commons Attribution License, which permits unrestricted use, distribution, reproduction and adaptation in any medium and for any purpose provided that it is properly attributed. For attribution, the original author(s), title, publication source (PeerJ) and either DOI or URL of the article must be cited.
License URL: https://creativecommons.org/licenses/by/4.0/

Keywords: Citrus pests, Beneficial species, Biodiversity, DNA metabarcoding, Biological control

Funding: Sino-America Biocontrol International Cooperation Program 59-0212-9-001-F This research was funded by Sino-America Biocontrol International Cooperation Program (59-0212-9-001-F). The funders had no role in study design, data collection and analysis, decision to publish, or preparation of the manuscript.

==============================
Background

DNA metabarcoding is rapidly emerging as a cost-effective approach for large-scale biodiversity assessment and pest monitoring. The current study employed metabarcoding to assess insect diversity in citrus orchards in Ganzhou City, Jiangxi, China in both 2018 and 2019. Insects were sampled using Malaise traps deployed in three citrus orchards producing a total of 43 pooled monthly samples.

Methods

The Malaise trap samples were sequenced following DNA metabarcoding workflow. Generated sequences were curated and analyzed using two cloud databases and analytical platforms, the barcode of life data system (BOLD) and multiplex barcode research and visualization environment (mBRAVE).

Results

These platforms assigned the sequences to 2,141 barcode index numbers (BINs), a species proxy. Most (63%) of the BINs were shared among the three sampling sites while BIN sharing between any two sites did not exceed 71%. Shannon diversity index (H′) showed a similar pattern of BIN assortment at the three sampling sites. Beta diversity analysis by Jaccard similarity coefficient (J) and Bray-Curtis distance matrix (BC) revealed a high level of BIN similarity among the three sites (J = 0.67–0.68; BC = 0.19–0.20). Comparison of BIN records against all those on BOLD made it possible to identify 40% of the BINs to a species, 57% to a genus, 97% to a family and 99% to an order. BINs which received a species match on BOLD were placed in one of four categories based on this assignment: pest, parasitoid, predator, or pollinator. As this study provides the first baseline data on insect biodiversity in Chinese citrus plantations, it is a valuable resource for research in a broad range of areas such as pest management and monitoring beneficial insects in citrus gardens.

Introduction

China ranks among the top three countries globally for citrus cultivation and production (Caserta et al., 2020). In fact, it ranked first in the world in both 2017 and 2018, producing 32 million tons of fresh citrus (CGA, 2020). Citrus fruit is considered a superior agricultural product and its production represents an important industry in the rural areas of southern China. However, crop yields are severely impacted by an array of pests. A large number of insect species, both pest and beneficial taxa (Niu et al., 2014) occur on citrus but their identification is difficult. Cryptic morphology and lack of taxonomic experts are the major challenges in large-scale insect diversity assessments.

DNA barcoding (Hebert et al., 2003), the characterization of sequence variation in a 658 bp fragment of cytochrome c oxidase I gene (COI), has gained global acceptance for specimen identification and species discovery (Gwiazdowski et al., 2015; Hebert et al., 2004). This method supplements morphology’s limitations by identifying unknown organisms by matching their barcode sequences to reference sequences. The simplicity and reliability of this method has motivated expansion of the DNA barcode reference library in the barcode of life data system (BOLD) (www.boldsystems.org). The BOLD system assigns eligible barcode sequences (>507 bp, <1% ambiguous bases, no stop codons, no contamination) to barcode index numbers (BINs) which serve as proxy for species (Ratnasingham & Hebert, 2013). BOLD currently holds seven million insect barcodes which have been assigned to more than 700,000 BINs. Implementation of the BIN system has enhanced the ability of DNA barcoding to discern and to count species (Hebert et al., 2016), to assess biodiversity composition (Ashfaq et al., 2018; Telfer et al., 2015), to map species distributions (Ashfaq et al., 2017), and to track species movements across borders (Ren et al., 2017). This success has led to the use of BINs in the analysis of bulk samples and biodiversity studies by high-throughput sequencing (HTS) (Cristescu, 2014).

DNA metabarcoding is a developing approach that identifies the species present in a mixed sample (bulk DNA or environmental DNA) based on HTS of a specific DNA marker (Comtet et al., 2015; Hajibabaei et al., 2011; Moriniere et al., 2016; Yu et al., 2012). It differs from conventional DNA barcoding (usually based on Sanger sequencing of individual specimens) because HTS allows taxonomy to be assigned to hundreds or even thousands of species in a bulk sample. It achieves this goal by generating amplicons of the barcode region from bulk DNA extracts which are then sequenced and assigned to operational taxonomic units (OTUs) that are queried against reference sequences to determine their source species (Cristescu, 2014). Metabarcoding is not intrinsically linked to the use of OTUs, but it could group the DNA reads in amplicon sequence variants (ASVs) (Callahan et al., 2016). Studies have now employed this approach to assess species composition in biological communities such as aquatic and terrestrial arthropods (Beng et al., 2016; Braukmann et al., 2019; Elbrecht & Steinke, 2018; Ji et al., 2013) making metabarcoding an increasingly cost-effective approach for large-scale biodiversity studies.

Widespread interest in metabarcoding has resulted in data proliferation and the development of computational tools to aid data analysis. As the BIN system has offered a novel approach to circumvent morphological bottlenecks to discriminate species, pairing of BINs with HTS has accelerated biodiversity assessments. Prior studies have used metabarcoding to survey insect diversity in different ecological settings in China (Huang et al., 2022), but reports on the use of this technology to explore insect diversity in fruit gardens in this region are lacking. The current study aimed to fill this gap by coupling metabarcoding with the BIN system to explore insect diversity in citrus plantations. The composition of insect communities in citrus orchards was analyzed using DNA metabarcoding followed by data analysis on BOLD and mBRAVE. The species revealed by BIN matches on BOLD were then searched in citrus pest database—citrus pest information system (CPIS, cpis.hzau.edu.cn) and the literature to allow their classification into pest, parasitoid, predator, or pollinator (Niu et al., 2014; Smaili, Boutaleb-Joutei & Blenzar, 2020; Urbaneja et al., 2020). The results provide a valuable resource for research on citrus pest management and beneficial insects. Exploration of pest and beneficial insect species in Chinese citrus orchards would allow scientists to screen the appropriate agents for pest management programs and promote pest biological control.

Materials and Methods

A single Malaise trap was deployed in three citrus orchards (GAN, QIU, SHI) about one kilometer apart in Ganzhou City, Jiangxi Province, China (Fig. 1). Samples were collected from April 2018 to July 2019 by replacing the collection bottles every month—so 43 samples were obtained from the three traps (Table S1). The collection bottles were stored at −20 °C until sorting. Specimens from each bottle were sorted into two size categories: small (e.g., parasitic wasps, flies) and large (e.g., butterflies, locusts). Large specimens were subsampled to obtain a tissue block (legs or partial abdomen) of a size similar to the small ones while small specimens were used in entirety. This was done to achieve comparable tissue-mass representation for all specimens in the DNA extracts. The specimens/tissue samples from each bottle were mixed, frozen in liquid nitrogen and then ground to a fine powder using a disposable mortar and pestle. DNA was extracted with the TIANamp Genomic DNA kit (DP304; TIANGEN Biotech, Beijing, China) following manufacturer’s protocols. Briefly, 500 mg of the powder was lyzed in 2 ml Buffer GA overnight at 56 °C in the presence of 20 µl proteinase K. The lysate was centrifuged at 5,000 g for 10 min and the supernatant was then aliquoted 40 µl into ten equal volumes for DNA extraction. The quality and purity of the DNA was assessed using a Nanodrop 2000 and 1% agarose gel electrophoresis. Three of the 10 DNA extracts from each sample were randomly selected for PCR, creating a total of 129 DNA extracts for analysis.

Figure 1 Three sites in Ganzhou (Jiangxi Province, China) where Malaise traps were deployed.

The map was created with GPSVISUALIZER (https://www.gpsvisualizer.com).

PCR amplification

A first PCR was used to amplify the target region of COI while the second PCR added adapters to allow discrimination of the sequences derived from each DNA extract (Prosser et al., 2016). The first PCR employed the following primer pair—AncientLepF3 (TTATAATTGGDGGWTTTGGWAATTG) (Prosser et al., 2016) and LepR1 (TAAACTTCTGGATGTCCAAAAAATCA) (Hebert et al., 2004) and the following thermocycling regime: initial denaturation for 5 min at 95 °C, then 25 cycles of denaturation for 30 s at 95 °C, followed by annealing for 30 s at 55 °C and extension for 60 s at 72 °C, and a final extension for 10 min at 72 °C. For the second PCR, adapters were added to primers employed in the first PCR. All the second PCR reactions had a total volume of 25 µl and included 12.5 µl 2×PCR Master Mix, 2 µl each of sequencing primers described above (10 µM), and 1 µl of template purified from the first PCR product or water (negative template control) and employed the following thermocycling regime (initial denaturation for 45 s at 98 °C, then six cycles of denaturation for 15 s at 98 °C, followed by annealing for 30 s at 60 °C and extension for 30 s at 72 °C, and a final extension for 1 min at 72 °C). Amplicons were purified using QIAquick Gel Extraction Kit and KAPA Library Quant (Illumina Inc., San Diego, CA, USA) and DNA Standards & Primer Premix were used for amplicon quantification. Satisfactory amplification was achieved from 126 of the 129 DNA extracts (Table S1).

High throughput sequencing (HTS)

Purified PCR2 products (126 reactions) were sequenced separately on an Illumina Miseq PE300 platform following standard protocols. Each reaction had two sequencing replicates to generate 252 products for sequencing. The amplicon libraries were prepared using MiSeq Reagent Kit v3 (Illumina Inc., San Diego, CA, USA). Briefly, the pooled library was thawed on ice along with HT1 (hybridization buffer), and then diluted to 2 nM in EB buffer. A total of 5 µl of the library were mixed with 5 µl of NaOH at 0.2 N in a microcentrifuge tube with brief vortexing and 1 min of centrifugation at 280 g. After 5 min incubation at room temperature, 990 µl pre-chilled HT1 was added to the tube containing denatured library providing 1 ml of a 10 pM denatured library. The denatured library was diluted to 8 pM (480 µl of the 10 pM denatured library, 120 µl of the pre-chilled HT1) by inverted mixing and then pulse centrifugation. Subsequently, the libraries were loaded onto the reagent cartridge to set up the sequencing run. After cleaning the flow cell, the reagents were loaded into the flow cell to initiate the sequencing.

Sequencing data analysis

There are a range of cloud-based platforms and tools that are freely available to analyze HTS data (Bani Baker et al., 2020). One such tool, mBRAVE, the multiplex barcode research and visualization environment, is a data storage and analytics platform with standardized pipelines and a sophisticated web interface designed to transform raw HTS data into biological insights (www.mbrave.net). mBRAVE integrates common analytical methods and links to the BOLD system for access to reference datasets (Young et al., 2021; Zieritz et al., 2022) and assignment of sequences to BINs, the features that are unique to this platform only. Results from the 252 sequence libraries were uploaded to mBRAVE (www.mbrave.net) under the project “MBR-MTCHN1” where they were analyzed using a standard pipeline (Young et al., 2021; Zieritz et al., 2022) involving sequence trimming (25 bp on each side), quality filtering (minimum QV 25), de-replication, identification, and OTU generation. mBRAVE has direct access to the DNA barcode reference libraries on BOLD which allows comparison of the sequence data with the selected libraries and interpretation of the outcome. This also allows the assignment of the generated OTUs to the barcode index numbers (BINs) and taxonomy on BOLD. The sequences were run against the DNA barcode reference library for Insecta on BOLD that represents 0.5 million BINs and 213,000 named species (the system reference library for mBRAVE ID engine—Insecta). The library also includes about 60,000 insect barcodes from China (DS-CHINAINS).

Sequences on mBRAVE were organized by ‘sequence runs’ and assigned to BINs and Linnaean taxonomy. The resultant data was downloaded from mBRAVE to summarize the results by Malaise trap location. Pairwise comparisons of BINs among trap locations were visualized by the package “VennDiagram” in R (ver. 4.2.2) statistical environment (R Core Team, 2023). To reduce the likelihood of false positives, a cleaning step was employed which excluded read counts in the BIN table that represented less than 0.01% of the total read count for their respective sample. BINs represented by a single sequence (singletons) were also excluded from the final BIN count. Concordance or discordance between a BIN and the associated species was determined using “BIN Discordance” tool on BOLD.

Diversity and species richness analyses were performed in R (package “vegan”) using BINs (as species proxies) recovered from the HTS sequence data. Alpha diversity at individual sites was analyzed by Shannon-Wiener index (Shannon, 1948) while the BIN similarity among the Malaise sites was calculated by Jaccard coefficient (Jaccard, 1912). Beta diversity was analyzed by Bray-Curtis dissimilarity index (Bray & Curtis, 1957). The species richness was assessed by three common estimators, Chao (Chao, 1984), Jack1 (first-order Jackknife) (Tukey, 1958) and bootstrap (Efron, 1992). Spread of BINs over time was determined by calculating BIN incidences in the Malaise sampling events for each site.

Results

The 252 HTS libraries included 14 Malaise trap collections from GAN, 15 from QIU and 14 from SHI (Table S1). In total, these runs yielded 9.5 million (M) DNA sequences which dereplicated to 2.7 M barcodes averaging 250 bp in length. The 1,515,627 sequences remaining after filtration were assigned to BINs. Most sequences belonged to Diptera (64%), Hymenoptera (16.1%), Lepidoptera (15.3%), Hemiptera (2.9%) and Coleoptera (1.3%) (Fig. 2). The BIN system (Ratnasingham & Hebert, 2013) linked the cumulative COI sequences from the three Malaise sites to a total of 2,141 BINs, and their counts were similar for the three orchards (GAN = 1,795; QIU = 1,792; SHI = 1,712) (Table 1). These BINs were used as a proxy for species to analyze insect biodiversity assemblages at the collection sites. Most (63%) of the 2,141 BINs were common among the three sampling sites, while only 6–8% were shared exclusively between site pairs, and 6% were unique to each site (Fig. 3). Alpha diversity analysis showed a similar Shannon index (H′) value (GAN = 6.9; QIU = 6.9; SHI = 6.8) for the three Malaise sites. Biodiversity overlap determined by Jaccard similarity coefficient (J) suggested a high level of BIN sharing among the three sites in pairwise comparisons (GAN-QIU = 0.67; QIU-SHI = 0.68; GAN-SHI = 0.68). Beta diversity assessed by Bray-Curtis index showed a low level of BIN dissimilarity in the pairwise site comparisons (GAN-QIU = 0.20; QIU-SHI = 0.19; GAN-SHI = 0.19). All samples combined yielded 2,141 BINs, but estimates range between 2,233 (Chao), 2,260 (Bootstrap) and 2,373 (Jack1). At all three sampling sites, most BINs (GAN = 769; QIU = 739; SHI = 751) were encountered only once in the 14/15 collection events and just a few (GAN = 12; QIU = 15; SHI = 17) were detected in all samples. However, the variance (R2) of BIN occurrence in the collection events at the three sampling sites remained low (Fig. 4).

Figure 2 Number of sequences assigned to 14 insect orders based on Malaise trap collections from three Chinese citrus orchards.

Table 1 Insect BINs recovered from three Malaise traps in citrus orchards and their assignment and discordance at different taxonomic ranks.

	Location	
	GAN	QIU	SHI	All locations	
Total BINs	1,795	1,782	1,712	2,141	
BINs assigned to order	1,784	1,776	1,707	2,128	
BINs assigned to family	1,744	1,739	1,664	2,076	
BINs assigned to genus	1,053	1,034	1,019	1,229	
BINs assigned to species	766	755	733	875	
BINs with family-level discordance	15	18	19	20	
BINs with genus-level discordance	85	87	89	98	
BINs with species-level discordance	220	218	216	235	

Figure 3 Venn diagram depicting the overlap in BIN composition among insect collections from three citrus orchards.

Figure 4 Incidence of BINs in the Malaise collections at three sampling cites, (A) GAN, (B) QIU and (C) SHI.

When the sequences were compared against all insect records on BOLD, 40% (875) of the BINs showed a match to a known species while 59% (1,229) were placed to a genus, 97% (2,076) to a family, and 100% (2,141) to an order. However, a check of the correspondence between the BINs and their associated taxonomy revealed 235 discordances at the species level, 98 at the genus level and 20 at the family level. At GAN site, 220 discordant BINs were found at the species, 85 at the genus and 15 at the family level. At QIU, 218 discordant BINs were found at the species, 87 at the genus and 18 at the family level. At SHI, 216 discordant BINs were found at the species, 89 at the genus and 19 at the family level (Table 1).

Most (96%) of the 2,141 BINs were linked to 14 insect orders, with a 49% to Diptera, 22% to Hymenoptera, 14% to Lepidoptera, 5% to Coleoptera, and 4.5% to Hemiptera. 875 of the 2,141 BINs were linked to a known insect species which included 443 pests, 223 pest/pollinators, 140 parasitoids, 52 predators, two predator/pollinators and 15 pollinators (Fig. 5 and Table S2).

Figure 5 Number of BINs associated with pest and beneficial insect species.

Discussion

Challenges of morphology and lack of taxonomic experts have limited the understanding of insect pest and natural enemy complexes in citrus orchards, compromising the efficacy of pest management tactics (Niu et al., 2014). The present study circumvented these limitations by employing DNA metabarcoding to assess insect diversity in three citrus plantations. The coupling of Malaise traps with metabarcoding has been successfully used to develop inventory for insect faunas in Sweden (Karlsson et al., 2020) and Germany (Moriniere et al., 2016). Malaise traps are useful for capturing flying insects and have been frequently used for barcode-based insect diversity assessments (Hardulak et al., 2020; Karlsson et al., 2020). DNA metabarcoding offers simultaneous identification of multiple species in bulk samples (Yu et al., 2012). Prior studies have used metabarcoding to examine insect biodiversity, pest prevalence (Piper et al., 2022), and to evaluate predator-prey relationships (Yang et al., 2022). For example, (Huang et al., 2022) used metabarcoding to reveal the composition of Diptera communities in a subtropical system, while (Kirse et al., 2021) used this method to analyze seasonal shifts in arthropod diversity in Malaise trap catches. The effective implementation of metabarcoding for simultaneous, multi-species identification of complex mixed communities have helped scaling up pest surveillance efforts (Piper et al., 2019).

BINs were used as a proxy to count species and to link the generated sequences to species represented on BOLD. This approach not only revealed the presence of potentially more than 2,000 insect species, but it also linked 866 (40%) of the BINs to known insects. This result supports the utility of BINs as an effective approach for counting species (Hebert et al., 2016) and assessing insect diversity (Telfer et al., 2015). While most BINs were assigned to single taxon, 12% were linked to more than one species or genus. The lower number of BINs with species matches on BOLD indicates the incompleteness of the current barcode reference library, emphasizing the need to develop regional DNA barcode reference libraries. BIN discordance has been reported in many barcode studies (De Leon et al., 2020; Gibbs, 2018) and the issue has generally been linked to misidentifications (Hebert et al., 2004), heteroplasmy, or incomplete lineage sorting (Kang et al., 2016; Weber, Stohr & Chenuil, 2019). These issues can only be resolved by detailed taxonomic analysis. Nonetheless, use of BINs to identify or count species in pooled samples where sequenced specimens may not be validated by morphology has limitations (Elbrecht et al., 2017).

The number of BINs revealed at each of the three sites were similar, but a majority (63%) of the BINs were common among sites, and this was reflected in the high alpha diversity (S = 0.67–0.68) and low beta diversity values (BC = 0.19–0.20). This means that there was little variation in the insect species at these locations. There are several possible reasons for the low insect beta diversity. One reason could be that due to a similar habitat of citrus plantations and small distance (1 km) between the three traps the physical characteristics of the habitat were relatively similar across a wide area, leading to a homogenization of insect species. For example, if the vegetation and climate are very similar across a region, or the geographic range is small, it may be more likely that the same insect species will be found throughout that region (Chesters et al., 2019; Dianzinga et al., 2020). To investigate the spread of BINs over time we calculated BIN incidences in the collection events. This analysis revealed that almost half of the BINs at each site were encountered only once in the total collection events at each site and just a fraction of them were detected in all samples. This was also supported by the species abundance analysis which estimated the presence of up to 2,273 BINs, more than 2,141 recovered from the three sites. Factors such as habitat loss, climate change, or use of pesticides may contribute to the loss or population fluctuation of certain insect species and role of these factors in our results cannot be ruled out (Forister, Pelton & Black, 2019). The collection of rare BINs in the traps may also point to the accidental visit of insects from the neighboring habitats, low species abundance, or seasonality of the activity of various insect species (Wolda, 1988).

Insects were classified as either pest or beneficial taxa based on the barcode sequence/BIN association with known species. The number of pest species detected was far higher than documented in the literature on citrus pests in China (Niu et al., 2014; Urbaneja et al., 2020). The present data provides a valuable resource for research in citrus pest management and monitoring. For example, the Asian citrus psyllid, Diaphorina citri, a devastating exotic pest of citrus in the US, was detected in samples collected from January to April, 2019. Its detection suggests the need for further monitoring targeted at this species to assist the timely implementation of control measures to reduce the risk of damage to the citrus production.

Numerous studies have been conducted to reveal the trophic relationships between predator and prey by detecting host DNA from the gut contents or feces of the predators using DNA barcoding (Furlong, 2015; Galan et al., 2018; McClenaghan, Nol & Kerr, 2019; Rytkonen et al., 2019; Verdasca et al., 2022). In our current study, we deployed the Malaise traps in the center of each citrus orchard, that reduced the possibility of adventitious predators. However, possibility of migration of insect predators from the neighboring crops to the sampling sites who ended up in the traps in search of their prey cannot be ruled out. Likewise, it is possible that some insects revealed in our study are not truly citrus pests instead they had been consumed by a predator or had moved to the gardens in search of nectar or to benefit from honeydew secreted by other arthropods. To minimize this deviation at most, we manually verified the identified insect species in the CPIS database and some published references described in materials and methods section, but contamination with non-target DNA originating from predator-prey interactions (Eitzinger et al., 2019) or airborne eDNA from off-site insects cannot be ruled out (Roger et al., 2022).

Biological control is the most environmentally safe and cost-effective pest management strategy. The present study revealed the presence of BINs linked to known species of Diptera and Hymenoptera which may have potential as biological control agents. For example, BINs linked to 32 species of Braconidae were encountered which belong to three genera (nine Apanteles, 14 Cotesia, nine Microplitis) that are potential biological control agents for Lepidoptera (Wharton, 1993). In addition, syrphid flies (48 species) revealed by BIN-linkages also can be useful natural enemies of aphids (Nelson et al., 2012).

In summary, metabarcoding of bulk insect collections provides a cost- and time-effective way to assess insect communities in citrus orchards. However, the present study only examined one region. Constructing a comprehensive insect inventory for citrus orchards across China will require surveys at the representative sites in other citrus areas.

Conclusions

Our study analyzed the composition of insect communities in citrus orchards using DNA metabarcoding followed by data analysis on BOLD and mBRAVE. The species revealed by BIN matches on BOLD were then searched in citrus pest database and the literature to allow their classification into pest, parasitoid, predator, or pollinator. Our results provide a valuable resource for research on citrus pest management and beneficial insect exploration.

Supplemental Information

Supplemental Information 1 Malaise collection samples and their PCR and Sequencing replicates.

Click here for additional data file.

Supplemental Information 2 Insect BINs and taxonomy recovered from Malaise traps deployed in three citrus orchards and their status as pest or beneficial insects.

Click here for additional data file.

We thank Changxiu Xia, Qinqin Wei and Yiwen Fang for the help with field sampling.

Additional Information and Declarations

Competing Interests

Author Contributions

Data Availability

The authors declare that they have no competing interests.

Chenxi Liu conceived and designed the experiments, performed the experiments, analyzed the data, prepared figures and/or tables, authored or reviewed drafts of the article, and approved the final draft.

Muhammad Ashfaq analyzed the data, prepared figures and/or tables, authored or reviewed drafts of the article, and approved the final draft.

Yanfang Yin performed the experiments, prepared figures and/or tables, and approved the final draft.

Yanjuan Zhu performed the experiments, prepared figures and/or tables, and approved the final draft.

Zhen Wang performed the experiments, prepared figures and/or tables, and approved the final draft.

Hongmei Cheng performed the experiments, prepared figures and/or tables, and approved the final draft.

Paul Hebert analyzed the data, prepared figures and/or tables, authored or reviewed drafts of the article, and approved the final draft.

The following information was supplied regarding data availability:

Citrus insect DNA metabarcoding data are available in NCBI SRA: PRJNA882198.

The data is available at Figshare: Liu, Chenxi (2022): Rawdata for insect DNA metabarcoding sequences in citrus in China. figshare. Dataset. https://doi.org/10.6084/m9.figshare.21153868.v1.

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
