# Peer review of "Using DNA metabarcoding to assess insect diversity in citrus orchards"

_PeerJ, doi:10.7717/peerj.15338_

## Round 0.1 · original submission · Major Revisions

The Manuscript has been critically reviewed by the subject experts and their review reports are attached. Although the reviewers appreciated the subject of the manuscript, they also highlighted the critical deficiencies in the Methodological section and inappropriate statistical analysis, which might lead to misleading findings. e.g. one review takes the point that time series of the trap could add more values. The authors have samples in the same areas for more than a year, which means that they should be able to show the seasonality of some of the insect species during this time frame. But this manuscript also provides useful baseline data for future studies. So according to the recommendations made by the reviewers, the manuscript needs a Major Revision.

Reviewer 1 ·

Basic reporting

The English language is clear, but I am not a native speaker myself.

In the introduction, the literature is well referenced and relevant with the exception of: when mentioning eDNA, please cite: TABERLET, P., COISSAC, E., HAJIBABAEI, M., and RIESEBERG, L.H. (2012), Environmental DNA. Molecular Ecology, 21: 1789-1793. https://doi.org/10.1111/j.1365-294X.2012.05542.x

In the introduction, you should explain the knowledge gap being filled. Please state more clearly the aims of this research. Also consider implementing lines 88–97 in the Material and Methods section rather than the Introduction.

The figures are relevant and of high quality. They are well labeled, with the exception of Figure 2. In Figure 2, are all the sequences included in the analysis or only those assigned to species? Because it is not clear from the figure description and y axis title. Furthermore, for Figure 1, please give more geographical context in the picture itself, showing where this part of China is on the map.

The structure includes all of the necessary components.

Experimental design

Please state your aims and the knowledge gap you are filling more clearly.
Methods should be described with more information as listed below.
For the collection method that you are describing in lines 100–103, please explain how you have ensured preservation of the DNA in the sample.
The methodology that you have described in Lines 119–134 lacks information about the use of positive as well as negative controls during DNA isolation and PCR amplification, to be able to detect instances of reagent contamination, primer and cross-contamination of samples. Please explain if you have used such controls in this method.
For the lines 119-120, could you please provide us with the full list of MID-tags that you have used?
For lines 137-138, they could specify whether the samples were pooled or sequenced separately.

Validity of the findings

For the subtitle "General patterns of biodiversity," you have not done any diversity analysis in this part, so please revise.
In order to provide further context for your findings in light of the manuscript's title, you should conduct diversity analysis between sites using alpha diversity (for example, the Shannon index) and similarity analysis (for example, the Jaccard index).
In the discussion in lines 229–231, you should cite work that has previously shown similar results (for example, De Leon, L.F.; Cornejo, A.; Gavilan, R.G.; Aguilar, C. Hidden biodiversity in Neotropical streams: DNA barcoding uncovers high endemicity of freshwater macroinvertebrates at small spatial scales. PLoS ONE, 15, e0231683, 2020.)
In lines 243-245, please cite more relevant papers.
It's unclear from the sparse writing on lines 243-249 whether you identify examples of prey species that the predators from your sample have eaten.

Additional comments

For the lines 161-161, please explain in more detail what you are showing with Figure S1 and why it is relevant.
In line 75 you mention eDNA abbreviation for the first time so consider writing full term.
For the lines 184–186, you should express these results as percentages, so it would be easier to read and understand.

I compliment the authors for their comprehensive data set, which was gathered over fieldwork. The manuscript is also written in a precise, formal style that avoids ambiguity. If there is a flaw, it is in the result analysis, which should be strengthened before Acceptance (as I mentioned before).

·

Basic reporting

The manuscript titled “Using DNA metabarcoding to assess insect diversity in citrus orchards” focuses on a very timely topic, the use of metabarcoding to assess insect diversity for Agriculture.
I think a work of this kind is very useful and can provide important information for researchers as well as farmers and agronomists.

The article is clearly written, and the background is well explained. Probably, more focus should be put on the differences between BINs as a proxy for species, and actual species records, since based on the results and discussion, the two concept are often confused.

The figures are useful to understand the text, although figure 3 may need to be redone (see comments below). I am not sure if the raw data has been shared, since no mention of it was made in the text.

Experimental design

The experimental design and the methods used are fit to the purpose, with three separate sampling areas in Citrus Orchards, each subsampled (3/10) at DNA extraction stage and at the sequencing step (2x), for a total of 43 samples, and 252 reps.

However, the results obtained do not seem to be fully used.
The authors initially state they want to analyse the composition of insect communities and differentiate them between categories (i.e., pests, beneficials). While the authors indeed obtained a list of species and classified them, this doesn’t provide much insight into the insect communities, since the samples were collected during a 15-month period of time. Many of these species may have never interacted with each other due to seasonality, making them hardly part of the same community.

Additional investigation should explore the community composition at a seasonal (monthly?) level.

Validity of the findings

The results obtained are valid and potentially useful, but they have not been explored sufficiently.

Furthermore, it seems the authors confuse the concepts of BINs as a proxy to species and an insect species in the strict sense. These are not the same thing and it should be made more clear in this work.

Please, see my comments below to more detailed issues.

Additional comments

Major corrections:
At lines 222-224 there is something quite confusing. It is true that BINs can be used as a species proxy, but it is also true that the authors have used the BINs to obtain species identification. The authors cannot say they have found more than 2000 insect species just because that’s the number of BINs they obtained. This is due to the fact that most of their BINs are also matching the same species. The authors managed to link a BIN to a species in 875 instances, but only 813 of these are unique instances (e.g., sometimes two BINs are linked to the same species). Even amongst these 813, there are instances of the same BIN assigned to multiple species. These are clear cases of taxonomical issues, where a species may be part of a complex or may have been recently revised.
This is one of the limitations of BINs. They can be used as a species proxy, but they are NOT the same as a species. The authors should decide if they want to stick to the BINs (in this case do not consider a BIN = a species), or if they want to go deeper in their analysis and focus on those BINs that could be identified to species.
Personally, I think that for this kind of work it would be more useful to get to a species-level. Describing how many species have been correctly identified.


Time series of the trap could add more values. The authors have samples in the same areas for more than a year, which means that they should be able to show the seasonality of some of the insect species during this time frame. It would add a lot of value to the manuscript if this could be incorporated into the discussion. For example, can the authors record associations between pests and parasitoids/predators across time? Are the parasitoids appearing after the pests? Can they see an increase in the number of parasitoid species where/when more pests are recorded?


Minor corrections:
Line 28: change “during” to “in”. It should read “in 2018-2019”.
Line 29: change sampled to “collected”. It should read “Insects were collected using..”
Line 30: change “collections” to “samples” and remove the parenthesis. It should read “43 pooled monthly samples.”
Line 101-103: In total there should be 45 samples. The supplementary table does not explain why samplers GAN and SHI had only 14 samples and QIU had 15. If all traps had a monthly collection for 15 months, there should be 45 samples. If samples were not collected/lost/failed, the authors should specify.
Line 110: it should be “manufacturer’s protocols”.
Line 119: not sure what the reference “Prosser et al. 2016” is doing there. If it refers to the use of MID, then it should go before the full stop, at line 121.
Line 127: remove “reaction” before “volume”. It should read “All PCR reactions had a total volume”.
Line 141: the number 5 should not be in letters.
Lines 171-174: these are very true information, but they are not methods. The authors should either move these lines to the introduction (when explaining the aims of the paper) or to the discussion.
Lines 181-182: The authors should remember that the overall number of traps they analysed was 43. This doesn’t change despite the number of DNA extraction subsamples and the number of sequencing reps. The numbers reported here are not correct/misleading. The Malaise trap collections are still 14 for GAN, 14 for SHI and 15 for QIU.
Figure 2: Where did the authors find the pictures of the insect? Can the authors ensure the figure for the Psocodea is actually correct? This looks like an Hemiptera hopper.
Figure 3: I personally find the figure slightly confusing. I think this is not how a Venn diagram should report the different amounts. The numbers reported in brackets are the TOTAL BINs for each orchard, these BINs should be subdivided between the various overlapping areas. The authors should remove the percentages and stick to the number of BINs. How many BINs are shared by all the orchards? How many shared only by two? Equally important (and currently completely missing from the figure, how many BINs are present only in one orchard?

Reviewer 3 ·

Basic reporting

no comment

Experimental design

no comment'

Validity of the findings

no comment'

Additional comments

Manuscript is a good contribution in the field of insect diversity assessment but i have given some suggestions which must be incorporated

Annotated reviews are not available for download in order to protect the identity of reviewers who chose to remain anonymous.

---

## Round 0.2 · Major Revisions

Overall the other two reviewers find merit in the revised submission. I am inclined to agree.

However, as suggested by Reviewer 2, the authors should perhaps consider whether the inclusion of Venn diagrams is the best method to show this result and perhaps revise the result presentation for the alpha diversity section.

Subsequently, for work of this nature( both in the organisms studied and the method of study with eDNA and it's workflow), the inclusion of beta diversity may or may not be possible. Since insects are very mobile, perhaps an inclusion of distance between the three sites can help alleviate concerns about differences in beta diversity.

Reviewer 1 ·

Basic reporting

Although the text is nicely organized, some of the figures and tables may use more clarification and improvment.

Figure 4- Location does not need to be stated on the figure; it can be stated in the figure description.
Table S1- Please be more specific about the information we can gather from the table.

The writing might need some consistency in certain areas of the text.

Line 28 The current study employed metabarcoding to assess insect diversity in citrus orchards in Ganzhou city, Jiangxi, China in 2018-2019 - Change to in both 2018 and 2019
Line 32 cloud-based data storage – Perhaps an expression data database would be preferable
Line 54- same as line 28
Line 59 Cryptic morphology and lack of taxonomic expertise are the major challenges in large-scale insect diversity assessments. – better to refer to the lack of taxonomic experts
Lines 63-64 This method circumvents the limitations of morphology by identifying unknown organisms by matching their barcode sequences to reference sequences - Perhaps it would be more accurate to argue that DNA barcoding supplements morphology's limitations.
Lines 139- adapt to the earlier text
Lines 142 and 150 – coordinate citing Illumina
Lines 222-226 - Clearly define percentages and full numbers, and make the writing consistent
Lines 201- NGS is a technique for obtaining sequences, not a particular kind of sequence; it is preferable to state the genetic marker and write COI sequences rather than NGS sequences.

Experimental design

The research question is clearly stated, and the methodologies used support it.

Validity of the findings

In comparison to the preceding version, conclusions were improved.

·

Basic reporting

This is the second revision I provide for this work.
As per my previous revision, the English writing is clear and unambiguous, with the literature references now expanded following the suggestions of some other reviewers.
I am generally hppy with the efforts the authors made to improve the work, however I think there is still a long way to go for this paper to be accepted on PeerJ.

Experimental design

The experimental desgin was good.

Validity of the findings

I have some major concerns on the following aspects:

1. Venn diagrams.

Unfortunately, Figure 3 is still very much incorrect, which I find a bit worrying.
First of all, I would not include the Jaccard results in the Venn Diagram. The authors should use the number of BINs.
Secondly, and most important, the this is not how a Venn diagram is presented: each intersecting space should have its own value (in this case a number of BINs), but for each site, the sum of values should be 100% of the diversity (total number of BINs for that site).
If the intersection of the three sites shows 63% of BINs in common, that means that each of the site should have a remaining % of BINs of 37% (100-63). So, the values currently included in the diagram that show a % of similarity between 73-75 (which are supposed to be the shared BINs between two sites) should be reported as 75-63=12% and 73-63=10%. Because these are the BINs in common between the two sites minus the BINs that are in common to all sites. Similarly, if site SHI (for example) has 63% of BINs share with the other sites, 10% shared only with GAN and 10% shared only with QIU, then it should have a total of 17% BINs present only in SHI (where you have included the H value).
The authors should use the number of BINs and, in bracket, the %.
For example, at the centre of the diagram, they should put the number of BINs shared by all sites: (this number needs to be calculated correctly, I am using here an approximation) ~1386 BINs (63%), at the intersection of GAN and SHI they should put ~220 BINs (10%), and so on. Remove from the Venn diagram both Jaccard and Shannon indexes, those can be reported/discussed in the results/discussion.

2. Alpha diversity measures.
I appreciate the use of Shannon and Jaccard diversity measures (for alpha-diversity), but:
- The authors should specify in the materials and methods what software they used to calculate the alpha diversity measures.

3. Beta diversity measures and testing.
Since the authors have calculated the alpha diversity, additional tests could explore the beta diversity across the different sites, including metrics calculating abundance, and phylogenetic distance. This would strongly benefit their work, allowing to determine if the different sites had a different beta diversity (were some species more present in some of the sites?). Adding phylogenetic metrics (such as Philr) would also allow the authors to determine if the insect diversity changed across sites based on the different insect groups (e.g., are there more dipteran insects in one of the sites?). This would add much more value to the work, since it would provide information on the composition of the different traps and the abundance of each BIN. Furthermore, some statistical test could be performed, in order to provide a better understanding of the significance of both alpha and beta diversity measures.

Additional comments

Additional comments:
- Use of HTS/NGS.
High throughput sequencing (HTS) and Next Generation Sequencing (NGS) are substantially synonyms. However, HTS has been recently replacing NGS (which is considered outdated). This is due to the fact that NGS was originally used to highlight the novelty of the technology, which has now been used for more than 15 years.. and it is not that novel anymore. The author should use consistently HTS and remove mention to NGS.

- Comments for the Discussion section.
As pointed out by other reviewers, it seems that the discussion is not really discussing in depth the finding of the work. This is a pity, since the finding are very valuable put are not put in any context. They are just listed, with no comments. A few examples:
- Lines 246-248: “Interestingly, the number of BINs revealed at each of the three sites were similar, but 25% of the BINs were unique to each site.” Why/how is this interesting? How does this result compare with other works? Would the author have expected a different result?
- Lines 248-249 (just following the comment above): “This result supports the utility of BINs as an effective approach for counting species (Hebert et al. 2016) and assessing insect diversity (Telfer et al. 2015).” How is this result supporting the utility of BINs? Discuss! If the authors had not used BINs, the results would have changed? I am not saying that BINs proved useful.. but it is not “this result” that supported the choice of using BINs.
- Lines 249-255: This is good! The authors have presented their results and discussed the findings in light of known/referenced works. This is how the whole discussion should be!
- Lines 258-262: If more than half of the BINs have been encountered only once, this is NOT seasonality. These are “rare species” or “low abundance” species. If it was seasonality, you would have larger populations at some time points of the sampling. For example, seasonality of some species is linked to the flowering of the plant, in Spring, and you would have recorded high number of individuals of these species during Spring vs a low number in Winter. But recording a single specimen across a two-years experiment, means that is not linked to the seasonality. So, why did the records show so many insects recorded only once? Where these species not associated with Citrus, windblown on the orchard?
- Lines 263-270: The authors went through all the effort of classifying their BINs as pests or beneficials but it seems they didn’t really use this data in their analysis. Does the percentage of pests/beneficial changes across sites? If yes, why do you think that is?

Reviewer 3 ·

Basic reporting

no comment

Experimental design

no comment

Validity of the findings

no comment

Additional comments

My suggested corrections have been added.

---

## Round 0.3 · Minor Revisions

Hello,

The reviewers have determined significant improvement in the current revised submission. Congratulations. However, some minor details remain to be addressed. Kindly attempt to address these as soon as possible.

Reviewer 1 ·

Basic reporting

If any graphs are not essential or not presented in the best way, the author should consider replacing or unifying them. To ensure that every detail is verified and listed in the most effective manner, the manuscript still has to be worked on. These are a few examples.

Figure 6 may just be black and white since the colors don't contribute any more information.
Figure 5: Is the equation required on the graph? See Albertini, A.; Pizzolotto, R.; Petacchi, R. Carabid patterns in olive orchards and woody semi-natural habitats: First implications for conservation biological control against Bactrocera oleae. BioControl 2017, 62, 71–83, for more information on how to perform instead rarefaction curves

Rephrase the statement if necessary, keeping in mind that you are referring to BINs throughout the text, not morphological species, when you mention Diptera species in lines 308–313 that are effective biocontrol agents.

Figure 4 doesn't seem to be particularly informative; in fact, the scale is not applied at all, and just two colors are visible; perhaps it should be substituted with a text sentence.

Experimental design

In comparison to the previous version, the experimental design is well explained.

Validity of the findings

The results are interesting, but there is room for improvement in the way they are presented.

·

Basic reporting

I am happy with the way the authors addressed all my previous comments and tried really hard to add some of the analyses I suggested.

Experimental design

The experimental design is well-explained and well-thought. I think the methods used are fitting well with the type of study presented.

Validity of the findings

The finding are useful for the research area they are meant for.

Additional comments

I have a few minor comments:

Line 37: this is not 8%, it is 63 (shared by all sites) + 8 (shared only by two sites) = 71%. The sentence should read “while BIN sharing between any two sites did not exceed 71%”.
Line 61: a verb is missing in this sentence.
Line 62: the name of the gene should be italicised. Please, italicise “cytochrome c oxidase”.
Lines 63-64: All these references are not needed. For this introductory sentence on DNA barcoding, the first work presenting DNA barcoding should suffice. Also, why the “standard fragment” is 658 bp? Not sure which reference you found this in, but I am not aware this is considered standard..?
Line 72: missing “s” in “assess”.
Line 73: for consistency, either use “to discern”, “to assess”, “to count” or remove the “to” from all of these.
Line 83: since you are explaining the general use of metabarcoding, you should mention that metabarcoding is not intrinsically linked to the use of OTUs, but could for example groups the DNA reads in amplicon sequence variants (ASVs). Check Piper at al. 2019 for an overview on this.
Line 109: there are two references after “a single Malaise trap”. Are these references describing a Malaise trap? Are these the first instance a Malaise trap was ever used? If not, then these references have no reason to be here since they are not adding value to the sentence. In order for a reference to add value to the sentence, this should either “explain” or “confirm” a statement.
For example, if the sentence had been “a single Malaise trap was deployed here since it has been proved to be an effective trapping system (deWaard et al. 2019)”, here the reference “confirms” your statement since the work of deWaard has used Malaise traps and proved them effective.
Line 113: change “used” to “sorting”.
Line 114: for consistency, use the plural everywhere. Change “locust” to “locusts”. And use the common names for everything, too. Change ”Drosophila” to fruit flies or flies.
Line 129: this sentence may benefit from some rewriting. Maybe use something like: “A first PCR was used to amplify the target region [….], while a second PCR added the adapters.
Line 131: “The first PCR..”. change all instances of PCR1/PCR2
Line 134 & 140: if you use the words “the following thermocycling regime” then do not use parenthesis but use a column.
Line 136: “For the second PCR..”.
Line 182: the reference for R software should always be with the current year. Please, change to (R Core Team 2023), also in the reference list.

Results:
- I don’t think the result section needs sub-headers, especially considering the “Screening pests and beneficial insect species” is very short. I would simply make this a separate paragraph within the “results” section.
- Not sure Figure 4 is really necessary. It doesn’t really show much and is probably superfluous once the beta diversity has been discussed in the text. An NMDS plot would be probably much better to show the low beta diversity.

Reviewer 3 ·

Basic reporting

no comment

Experimental design

no comment

Validity of the findings

no comment

Additional comments

Manuscript is written well and has signification contribution towards recent tools for insect identification.

---

## Round 0.4 · accepted · Accept

Congratulations on your submission being accepted for publication.

Reviewer 1 ·

Basic reporting

Since the initial draft, the manuscript has been improved, and the results and methodology are now presented in a clearer manner.

For instance, the text only need a few minor corrections:
Lines 211-213: the text in brackets is in italics, is that necessary?
Figure 4: try to remove -100/-200 from the y axis, it's confusing

Experimental design

Without additional comments.

Validity of the findings

Without additional comments.

Additional comments

Without additional comments.

·

Basic reporting

The authors have addressed all my comments in regards to basic reporting.

Experimental design

The authors have addressed all my comments in regards to experiemental design.

Validity of the findings

The authors have addressed all my comments in regards to the validity of the findings. I remain of the idea this dataset could have been probably explored/mined further to better undetstand the role of all the predators/parasitoids/pollinators in relation to the pests of citrus. Hoewever, I think that the work in its current form still presents data and results that are useful to the scientific community.

Additional comments

The authors have addressed all my comments and I think the manuscript is now ready for publication.